# FOXN3 and GDNF Polymorphisms as Common Genetic Factors of Substance Use and Addictive Behaviors

**DOI:** 10.3390/jpm12050690

**Published:** 2022-04-26

**Authors:** Andrea Vereczkei, Csaba Barta, Anna Magi, Judit Farkas, Andrea Eisinger, Orsolya Király, Andrea Belik, Mark D. Griffiths, Anna Szekely, Mária Sasvári-Székely, Róbert Urbán, Marc N. Potenza, Rajendra D. Badgaiyan, Kenneth Blum, Zsolt Demetrovics, Eszter Kotyuk

**Affiliations:** 1Department of Molecular Biology, Institute of Biochemistry and Molecular Biology, Semmelweis University, 1094 Budapest, Hungary; vereczkei.andrea@med.semmelweis-univ.hu (A.V.); andrea.belik@gmail.com (A.B.); sasvari.maria@med.semmelweis-univ.hu (M.S.-S.); 2Institute of Psychology, ELTE Eötvös Loránd University, 1075 Budapest, Hungary; magi.anna@ppk.elte.hu (A.M.); fajudit@gmail.com (J.F.); eisinger.andrea@ppk.elte.hu (A.E.); kiraly.orsolya@ppk.elte.hu (O.K.); szekely.anna@ppk.elte.hu (A.S.); urban.robert@ppk.elte.hu (R.U.); kotyuk.eszter@ppk.elte.hu (E.K.); 3Doctoral School of Psychology, ELTE Eötvös Loránd University, 1075 Budapest, Hungary; 4Nyírő Gyula National Institute of Psychiatry and Addictions, 1135 Budapest, Hungary; 5International Gaming Research Unit, Psychology Department, Nottingham Trent University, Nottingham NG1 4FQ, UK; mark.griffiths@ntu.ac.uk; 6Departments of Psychiatry, Child Study and Neuroscience, Yale University School of Medicine, New Haven, CT 06511, USA; marc.potenza@yale.edu; 7Connecticut Council on Problem Gambling, Wethersfield, CT 06109, USA; 8Connecticut Mental Health Center, New Haven, CT 06519, USA; 9Department of Psychiatry, Ichan School of Medicine at Mount Sinai, New York, NY 10029, USA; rbrb@buffalo.edu; 10Division of Addiction Research & Education, Center for Psychiatry, Medicine, & Primary Care (Office of the Provost), Western University Health Sciences, Pomona, CA 91766, USA; drdgene@gmail.com; 11Centre of Excellence in Responsible Gaming, University of Gibraltar, Gibraltar GX11 1AA, Gibraltar

**Keywords:** addictive behaviors, genetic association analysis, substance use, substance-related disorders, *FOXN3*, *GDNF*

## Abstract

Epidemiological and phenomenological studies suggest shared underpinnings between multiple addictive behaviors. The present genetic association study was conducted as part of the Psychological and Genetic Factors of Addictions study (*n* = 3003) and aimed to investigate genetic overlaps between different substance use, addictive, and other compulsive behaviors. Association analyses targeted 32 single-nucleotide polymorphisms, potentially addictive substances (alcohol, tobacco, cannabis, and other drugs), and potentially addictive or compulsive behaviors (internet use, gaming, social networking site use, gambling, exercise, hair-pulling, and eating). Analyses revealed 29 nominally significant associations, from which, nine survived an FDRbl correction. Four associations were observed between FOXN3 rs759364 and potentially addictive behaviors: rs759364 showed an association with the frequency of alcohol consumption and mean scores of scales assessing internet addiction, gaming disorder, and exercise addiction. Significant associations were found between GDNF rs1549250, rs2973033, CNR1 rs806380, DRD2/ANKK1 rs1800497 variants, and the “lifetime other drugs” variable. These suggested that genetic factors may contribute similarly to specific substance use and addictive behaviors. Specifically, FOXN3 rs759364 and GDNF rs1549250 and rs2973033 may constitute genetic risk factors for multiple addictive behaviors. Due to limitations (e.g., convenience sampling, lack of structured scales for substance use), further studies are needed. Functional correlates and mechanisms underlying these relationships should also be investigated.

## 1. Introduction

Psychological and biological overlaps between different types of addictions have long been noted. At the phenomenological and diagnostic levels, the American Psychiatric Association (APA) defines substance use disorders as complex conditions “in which there is uncontrolled use of a substance despite harmful consequence.” The APA also states that individuals can develop addictions not only to substances but to different behaviors, such as gambling. The classification and the diagnostic criteria of addictive behaviors in the fifth edition of the *Diagnostic and Statistical Manual for Mental Disorders* ([1] and the eleventh revision of the *International Classification of Diseases* [2] also reflect phenomenological similarities of addictive behaviors. The DSM-5 includes the updated category of “Substance-related and Addictive Disorders” replacing the former “Substance-related Disorders” category from the DSM-IV-TR. Although only gambling disorder is officially included in the DSM-5 (under the Non-Substance-Related Disorders category of Substance-Related and Addictive Disorders; internet gaming disorder was included in Section 3 as a tentative disorder requiring more research before official inclusion), the new “Substance-related and Addictive Disorders” terminology in the DSM-5 reflects a formal acknowledgment of behavioral addictions that is based on empirical data [3]. Furthermore, in ICD-11, gambling and gaming disorders are both included and classified as disorders due to addictive behaviors [4,5,6].

Co-occurrences and comorbidities are common, as many individuals with addiction-related diagnoses have more than one psychiatric diagnosis [7]. In a large US-population-based study, 96% of individuals with gambling disorder experienced one or more co-occurring psychiatric disorders, with DSM-IV nicotine dependence, substance abuse, and substance dependence found in over 60%, 40%, and 30%, respectively ([8,9]. The co-occurrence of substance and behavioral addictions is well documented in other studies [10,11,12,13,14,15,16,17,18].

As for the neurobiological systems involved in addiction, the basal ganglia, the ventral tegmental area, the striatum [19], and the prefrontal prelimbic cortex have been implicated in previous studies [20,21]. Furthermore, these neurocircuits interact with other circuits, such as motivational, stress, and mood regulation circuits, involving the hypothalamus, amygdala, and habenula [22]. The main neurotransmitter is dopamine, showing a large and fast increase related to feelings of being “high” in, for example, stimulant use disorders [23,24]. Furthermore, opioid peptides, serotonin, acetylcholine, GABA (gamma aminobutyric acid), glutamate, endocannabinoid systems, orexin, and corticotropin-releasing factor have also been implicated [20,21].

These systems were the focus of multiple studies on substance abuse, non-substance addictions, and various risk tendencies, such as novelty seeking, impulsivity, or aggressive behaviors; see reviews from our group [25,26,27] and others [28,29,30]. For example, common molecular pathways of substance and behavioral addictions were suggested between cocaine and alcohol addictions and compulsive running [31,32]. Overall, neurobiological research suggested that different substances and behaviors may impact both similar and distinct neurobiological pathways [26,27,33,34].

From a genetic perspective, family, twin, and adoption studies estimated the heritability (i.e., the overall genetic contribution) of addictions to be 30–70% [35,36,37,38]. Studies also suggested genetic overlaps between behavioral (e.g., gambling) and substance (e.g., alcohol, tobacco, cannabis, and stimulant) addictions [39,40,41,42]. Furthermore, the genetic and environmental overlap between different types of substance and behavioral addictions has long been suggested [43,44]. Although most genetic studies tend to separately address substance abuse (usually focusing on a single substance) and behavioral addictions, there are some studies available that examined multiple types of addictions. Twin (and other) studies suggested that most shared genetic and environmental factors may not be substance-specific [45,46,47,48,49], although genome-wide association studies often implicate genes involved in substance metabolism and subjective responses to specific drugs [50,51]. Identified genetic variants in addiction-related genes (e.g., aldehyde dehydrogenases [*ALDHs*], Gamma-Aminobutyric Acid Type A Receptor Alpha2 Subunit [*GABRA2*], and DRD2/Ankyrin repeat and kinase domain containing 1 [*ANKK1*]) were linked to dependence on various substances [52,53,54,55]. Gene network analysis showed immune signaling and extracellular signal-regulated protein kinases 1 and 2 (*ERK1/2*) as novel genetic markers for multiple addiction phenotypes including alcohol, tobacco, and opioid use disorders [56]. Moreover, shared genetic contributions to gambling and substance use disorders were described [39,40,41,57,58].

Twin research has observed a genetic overlap between substance use and gambling frequency [59]. A genome-wide association study (GWAS) provided a linked polygenic risk score for alcoholism and gambling disorder [60], although no findings from this and another GWAS identified any region that reached genome-wide significance for gambling disorder [61]. Some common allelic variants were also implicated in personality traits, such as risk-taking, which were linked to alcohol and drug use [62]. As such, some research findings suggested that specific genetic markers may increase the likelihood of addictions generally. Pharmacological studies have shown treatment non-specificity, suggesting similar underlying neurobiological pathways across addictions [63,64,65,66]. The theory of addiction as a substance-independent disease is further supported by data that individuals recovering from a given substance may switch one substance or behavior for another (e.g., from opiates to cocaine, alcohol, gambling) before successfully recovering from addiction [67,68,69,70].

In summary, co-occurrences of different types of addictive behaviors, twin studies, similarities in phenomenological and behavioral characteristics, and empirical studies of psychological and molecular mechanisms suggest commonalities across substance and behavioral addictions. The aim of the present study was to investigate possible genetic overlaps between different types of substance-related, addictive, and compulsive behaviors utilizing the large sample of Psychological and Genetic Factors of Addictions (PGA) study [71]. The uniqueness of this study is the genetic analysis of both substance and behavioral addictions within the same cohort. The sample was assessed for a wide range of potentially addictive substances (nicotine, alcohol, cannabis, and other drugs) and potentially addictive behaviors (internet use, gaming, social networking site use, gambling, exercising, hair-pulling, and eating). The present genetic association analysis was conducted by focusing on 32 single-nucleotide polymorphisms (SNPs) in candidate genes that were formerly implicated in studies of various addictions.

## 2. Materials and Methods

### 2.1. Data Collection and Analyzed Phenotypes

The present genetic association analysis was conducted as part of the PGA Study [71]. Data were collected between 2011 and 2015 at Hungarian high schools, colleges, and universities. In total, 3003 young adults participated in the study (mean age was 21 years; SD ±2.8 years). All participants signed written informed consent, provided buccal samples, and were administered self-report questionnaires. Participation was anonymous, and questionnaire data and DNA information were paired using a unique identification number for each participant. The study protocol was designed in accordance with the guidelines of the Declaration of Helsinki and was approved by the Scientific and Research Ethics Committee of the Medical Research Council (ETT TUKEB).

Substance consumption/use behaviors were assessed using questionnaires. Since the aim of the present genetic association analysis was to explore the genetic background of addictions, we included “never” users and regular users of specific substances. Smoking (tobacco) use was assessed with the question, “Do you smoke regularly or occasionally?”, with possible responses as “never”, “occasionally”, and “regularly”. Genetic association analysis was conducted on the “never” and “regular” cigarette-smoking groups (excluding individuals who occasionally smoked from analyses). Alcohol consumption was assessed with the question, “How many times did you drink 6 or more alcoholic drinks in the past 30 days?”, with possible answers being “never”, “1–3 times”, “4–9 times”, “10–19 times”, “not every day, but 20 or more times”, and “every day”. In order to define the regular alcohol user group, participants drinking 4–9 or more times in the past 30 days were categorized as alcohol-using. Cannabis consumption was assessed with the question, “How often did you consume marijuana in the past 30 days?”, with possible responses of “I did not use marijuana in the past 30 days”, “1–3 times”, “4–9 times”, “10–19 times”, “not every day, but 20 or more times”, and “every day”. In order to define the regular-marijuana-consuming group, participants consuming marijuana 4–9 or more times in the past 30 days were categorized as cannabis-using. With regard to other drugs (synthetic marijuana, amphetamine, cocaine, heroin, lysergic acid diethylamide [LSD], magic mushrooms, gamma hydroxybutyrate [GHB], mephedrone, steroids, alcohol with drugs, sedatives, other drugs), positive responses were rare; for detailed prevalence data see [14]. Therefore, a “lifetime other drugs” variable was created comprising participants who had ever used any of these drugs versus those who had never used any of them. In this classification of the “lifetime other drugs”, missing answers were excluded. However, since there was a high rate of missing data regarding synthetic marijuana (the number of missing answers was 1280), those who did not answer this question but answered the questions regarding the use of all other substances were also included in the appropriate never or ever other drug use groups.

With regard to behavioral addictions, they were assessed using the Problematic Internet Use Questionnaire [72] (PIUQ), Problematic Online Gaming Questionnaire Short-Form [73] (POGQ-SF), Bergen Social Media Addiction Scale [74,75] (BSMAS), Exercise Addiction Inventory [76,77] (EAI), Massachusetts General Hospital Hairpulling Scale [78] (MGH-HPS), Diagnostic Statistical Manual-IV-Adapted for Juveniles gambling criteria [79] (DSM-IV-MR-J), and the SCOFF eating disorders questionnaire [80]. For further details on the study design, materials, and participant description, see [71]. The data that support the findings of this study are available from the corresponding author upon request.

### 2.2. SNP Selection Criteria

Due to power considerations based on the sample size, as well as technical limitations of the genotyping apparatus at our institution (see below), we decided to concentrate our association study on 32 single-nucleotide polymorphisms (SNPs). Genes and their polymorphisms were selected from earlier GWASs and candidate gene association studies in the literature to best represent the expected distribution pattern of the various phenotypes assessed in the sample [81,82,83,84]. For example, nicotinic acetylcholine receptor gene clusters implicated by earlier GWASs [85,86,87] or polymorphisms proposed in the Genetic Addiction Risk Severity (GARS) by [88], along with some conventional receptors and metabolizing enzymes, were selected and a few novel genetic targets were also considered (see Appendix A).

### 2.3. DNA Preparation and SNP Genotyping

A non-invasive DNA sampling method using buccal swabs was utilized to obtain a sufficient amount of buccal cells for the genetic analysis, followed by DNA isolation (for a slightly modified method, see [89]). The swabs were incubated overnight at 56 °C in a lysis solution containing 0.2 g/L Proteinase K, 0.1 M NaCl, 0.5% SDS, and 0.01 M pH = 8 Tris buffer. After an RNase treatment, proteins were removed via salting out with saturated NaCl solution. DNA precipitation with isopropanol and ethanol was followed by the resuspension of the samples in 5 mM pH = 8 Tris and 0.5 mM EDTA. The concentration was measured using a NanoDrop 1000 Spectrophotometer (Thermo Fisher Scientific, Waltham, MA, USA). A DNA concentration range of 15–300 ng/µL was considered sufficient for further analysis and samples with lower concentrations were excluded.

Genotyping was performed using the Applied Biosystems™ QuantStudio™ 12K Flex system (Thermo Fisher Scientific). This high-throughput method is based on the application of pre-designed TaqMan^®^ fluorescent probes that are highly specific for the variable sequence of interest, which is immobilized on an array along with target-specific primers. The reaction mixture consists of the DNA sample (approximately 30–150 ng) and genotyping master mix (each deoxyribonucleotide triphosphate and the AmpliTaq Gold^®^ DNA polymerase, provided by the manufacturer) was loaded on the genotyping plates by the Applied Biosystems™ OpenArray™ AccuFill™ System (Thermo Fisher Scientific). Following the endpoint detection, the allele-specific FAM and VIC fluorescent intensities were evaluated using the QuantStudio 12 K Flex Software and the Thermo Fisher Cloud service. The Open Array format enables simultaneous genotyping of 32 SNPs on 96 samples on a single chip. Approximately 10% of the samples were measured in duplicates on the OpenArray system for quality control, where the reproducibility was higher than 98%.

The genotype distribution, as well as the Hardy–Weinberg equilibrium, and call rate data for the studied 32 SNPs are presented in Appendix A. Genotype distribution showed a significant deviation from the theoretically expected Hardy–Weinberg equilibrium in the case of *FOXN3* rs759364, *CNR1* rs2023239, and *CHRNA5/A3* rs1051730 variants. The observed genotype distributions of all other SNPs were in Hardy–Weinberg equilibrium.

### 2.4. Statistical Analysis

Genetic association analysis was conducted using an allele-wise design. In the case of the genetic association analysis of substance-use behaviors, allele frequencies of using and non-using groups (as defined in Section 2) were compared using Chi-square tests. In the case of potentially addictive behaviors, mean scores of the scales were compared using the alleles of the 32 SNPs. The number of participants slightly differed across the analysis based on the call rate of the genotypes. For the correction for multiple testing, the FDRbl adjustment of false discovery rate correction was assessed [90,91], where the level of significance for 32 tests was *p* < 0.01221.

## 3. Results

### 3.1. Descriptive Statistics of Substance Use and Behavioral Addiction Measures

Of the 3003 participants, 59.7% never smoked cigarettes (*n* = 1794) and 17.1% regularly smoked cigarettes (*n* = 513). Regarding alcohol consumption, 60.2% did not consume (*n* = 1769) and 7.3% reported drinking six or more alcoholic drinks in the past 30 days four or more times (*n* = 220). Regarding cannabis use, 70% did not use (*n* = 2102) and 2.7% reported using cannabis in the past 30 days four or more times (*n* = 80). Descriptive statistics of the behavioral addiction scales are described in detail elsewhere [14].

### 3.2. Genetic Association Analysis

The genetic association analysis results are presented in Table 1 for potentially addictive substances and Table 2 for potentially addictive behaviors. Twenty-nine nominally significant associations were observed, from which, nine associations remained significant after correcting for multiple testing. Regarding substance use associations, five remained significant after correcting for multiple testing (Table 1). The association between *FOXN3* rs759364 and an alcohol consumption frequency of six or more times a month was significant (χ^2^ = 9.116, df = 1, *p* = 0.0025), where the A allele increased the odds of alcohol consumption frequency of six or more times a month (OR = 1.34 [1.11–1.63]) compared to the G allele. The minor allele (A) was more frequent in the using (37.5%, *n* = 147) as compared to the non-using (30.0%, *n* = 935) group. The number of participants in the G allele using group was 246 and the number in the G allele non-using group was 2179. The analysis also showed a significant association between *DRD2/ANKK1* rs1800497 and consuming cannabis at least four times during the past 30 days (χ^2^ = 6.424, df = 1, *p =* 0.0113), where the A allele increased the odds of cannabis consumption frequency of four or more times a month (OR = 1.60 [1.11–2.30]). The association showed that the minor allele (A) was more frequent among the using (27.1%, *n* = 38) as compared to the non-using (18.6%, *n* = 699) group. The number of participants in the G allele using group was 102, and the number in the G allele non-using group was 3059. The chi-square analysis also showed a significant difference between the allele frequencies of *GDNF* rs1549250 for any other drug use (χ^2^ = 8.845, df = 1, *p* = 0.0029, OR = 0.85 [0.77–0.95]). The minor allele (C) was more frequent in the using (46.0%, *n* = 502) as compared to the non-using (40.9%, *n* = 1474) group. The number of participants in the A allele using group was 590, and the number in the A allele non-using group was 2130. An association was also observed between *GDNF* rs2973033 and drug use (χ^2^ = 7.060, df = 1, *p* = 0.0079, OR = 1.16 [1.04–1.30]). The minor (C) allele was more frequent among the using (31.5%, *n* = 343) as compared to the non-using (27.3%, *n* = 980) group. The number of participants in the T allele using group was 747, and the number in the T allele non-using group was 2606. Drug use also showed an association with *CNR1* rs806380 (χ^2^ = 7.095, df = 1, *p* = 0.0077, OR = 0.87 [0.78–0.96]). The minor allele (G) was more frequent among the using (36.4%, *n* = 407) as compared to the non-using (32.1%, *n* = 1173) group. The number of participants in the A allele using group was 711, and the number in the A allele non-using group was 2479.

When comparing the mean scores of the behavioral addiction scales in ANOVAs, four associations survived correction for multiple testing (Table 2). The *FOXN3* rs759364 showed a significant association with scores on the Problematic Internet Use Questionnaire (F[1, 5214] = 12.846, *p* = 0.0003, Cohen’s *d* = 0.11). The minor allele was associated with higher mean scores on the PIUQ: 10.53 ± 3.67 (*n* = 1594) vs. 10.13 ± 3.63 (*n* = 3622). The *FOXN3* rs759364 also showed a significant association with scores on the Problematic Online Gaming Questionnaire (F[1, 5000] = 8.788, *p* = 0.0030, Cohen’s *d* = 0.09). The minor allele was associated with higher mean scores on the POGQ: 16.11 ± 6.40 (*n* = 1552) vs. 15.55 ± 6.05 (*n* = 3450). Therefore, in both cases, the *FOXN3* rs759364 A allele was associated with higher mean scores on the scales. *FOXN3* rs759364 also showed a significant association with scores on the Exercise Addiction Inventory (F[1, 5208] = 9.105, *p =* 0.0026, Cohen’s *d =* 0.09) with a higher mean score for the major allele (12.63 ± 5.04, *n* = 3612) as compared to the minor allele (12.18 ± 4.83, *n* = 1598). A significant association was also found between eating disorder scores and *DRD4* rs1800955 (F[1, 4972] = 9.184, *p* = 0.0025, Cohen’s *d* = 0.09), where the minor allele was associated with lower mean scores on the EAI (0.67 ± 0.92, *n* = 2298) as compared to the major (T) allele (0.76 ± 0.98, *n* = 2676).

## 4. Discussion

Previous candidate gene and genome-wide association studies of addictions have mostly focused on investigating possible relationships between genetic variants and one specific type of substance use disorder or behavioral addiction. The present study examined the possible associations between 32 polymorphisms and a wide spectrum of substance and non-substance addictions in a large sample consisting of high school and university students to identify possible common genetic factors underlying addictions.

The genetic association analysis presented here was conducted as part of the PGA study [71] assessing multiple addictive behaviors in 3003 young adult participants. The associated analyses of 32 SNPs regarding the use of four potentially addictive substances (alcohol, tobacco, cannabis, and other drugs) and engagement in seven potentially addictive behaviors (internet use, gaming, social networking sites use, gambling, exercising, hair-pulling, and eating) found 29 nominally significant associations, from which, 9 remained significant after an FDRbl correction for multiple testing.

Four of the nine significant associations were observed between a *FOXN3* SNP and various addictive behaviors: rs759364 showed an association with the frequency of alcohol consumption and the mean behavior addiction scores (based on internet addiction, gaming addiction, and exercise addiction scales). The genotype frequencies of the *FOXN3* rs759364 SNP observed in the present sample showed a significant deviation from the expected genotype frequencies according to the Hardy–Weinberg equilibrium (Appendix A). This deviation from the theoretically expected frequencies was probably due to the low frequency of the AA genotype, but further studies are needed to verify this association. Forkhead box protein N3 (*FOXN3*) is a member of the forkhead/winged-helix transcription factor family and has been shown to act both as a transcriptional repressor [92,93,94,95] and activator [96]. Forkhead transcription factors belong to one of the major transcription factor families in eukaryotes and contribute to development, immunity, metabolism, and cell cycle control [97]. Along with the transcription repressor role, interaction with histone deacetylase complexes involved in the DNA damage response was also shown [92,93,95], as well as possible gene regulatory functions [98].

*FOXN3* has been previously associated with suicidal behavior in a GWAS [99] and the expression of *FOXN3* mRNA was also increased in the brains of individuals who had committed suicide [100,101]. Behavioral and molecular genetic studies have demonstrated the heritability of suicide behavior alongside substance abuse phenotypes [102]. The presence or absence of alcohol use was previously associated with suicide-specific genes in Polish populations [103]. The rs759364 SNP is an intronic variant of *FOXN3*, which has been associated with alcoholism and cigarette smoking [104]. Quantitative linkage analyses of 1717 SNPs showed a linkage peak for cannabis dependence on chromosome 14 bounded by rs759364 [105]. This peak includes candidate genes, such as *SERPINA1* and *SERPINA2* (serine-peptidase inhibitor, clade A, members 1 and 2), which were previously implicated in a GWAS for substance abuse vulnerability [106]. An association between *FOXN3* expression and areca nut chewing habits in patients with oral cancer has also been shown [107].

In the present study, we found two associations between *GDNF* variants: the intronic rs1549250 and rs2973033 in the 5′ untranslated region of the gene) and the “lifetime other drugs” variable. Glial-cell-line-derived neurotrophic factor (GDNF) plays a vital role in peripheral neuronal development [108], including axon guidance [109]. GDNF is produced by striatal neurons [110] and transported via dopaminergic neurons to the substantia nigra [111] and ventral tegmental area (VTA) [112]. GDNF positively regulates dopaminergic activity in nigrostriatal and mesolimbic projections [112,113].

Drugs of abuse can affect the expression of neurotrophic factors. Manipulation of neurotrophic factor levels can modify drug-seeking behavior [114]. The administration of GDNF in the mesocorticolimbic system has been linked to increased craving [115]. In contrast, several alcohol studies showed acute inhibitory effects of GDNF on drug-seeking behavior [116,117]. Upregulation in endogenous levels of *GDNF* mRNA and protein was found in rat VTA ten hours after a single administration of 20% ethanol, while no change in the nucleus accumbens was observed, suggesting that the ethanol-mediated effects on *GDNF* expression may be restricted to the VTA [118]. *GDNF* expression and downstream signaling were also found to be modulated by cocaine and amphetamines [119].

Reduced *GDNF* expression may also potentiate methamphetamine self-administration, enhance motivation to take methamphetamine, increase vulnerability to drug-primed reinstatement, and prolong cue-induced reinstatement of extinguished methamphetamine-seeking behavior [120]. Previous studies have associated *GDNF* polymorphisms with methamphetamine use [121] and cigarette smoking [122]. The rs2973033 variant of *GDNF* has been associated with a culturally distinct form of gambling in an Indian population [123].

The “lifetime other drugs” variable has also been associated with two additional polymorphisms: the rs806380 variant of the Cannabinoid Receptor 1 (*CNR**1*) gene and rs1800497 of the *DRD2*/Ankyrin Repeat and Kinase Domain Containing 1 (*ANKK1*) gene. *CNR1* codes for cannabinoid 1 receptors that showed high expression in brain regions that were linked to reward, addiction, and cognitive function [124,125]. The rs806380 SNP in intron 2 of *CNR1* has been associated with cannabis dependence [126]. However, it should be noted that most participants also met the criteria for alcohol dependence. The rs806380 SNP was also associated with the development of cannabis dependence symptoms [127]. A microsatellite polymorphism of *CNR1* has also been positively associated with intravenous drug use and cocaine, amphetamine, and cannabis dependence [128]. The rs1800497 SNP of *ANKK1*, also known as the *Taq*IA polymorphism, was widely studied among individuals with psychiatric disorders. Initially associated with the gene coding for the dopamine D2 receptor, the *Taq*IA polymorphism has been linked to reduced dopamine D2 receptor densities and binding affinity [129,130,131,132]. However, subsequent studies observed linkage disequilibrium between *DRD2* and *ANKK1*, with some studies linking associations with substance use behaviors more to *ANKK1* than *DRD2* [26]. Therefore, there may be direct or indirect influences on the concentration of dopamine in the synaptic clefts. The *Taq*IA polymorphism has been associated with nicotine dependence [133,134], cigarette smoking cessation [135], alcohol use disorder [136,137], opioid dependence [138,139,140], and cocaine dependence [141,142].

Finally, we found an association between rs1800955 of the *DRD4* gene and the mean scores on the SCOFF questionnaire assessing eating disorders. This SNP, also known as -521 C/T, is a variant in the promoter region upstream of the *DRD4* gene and has a putative role in the regulation of transcriptional activity [143]. A previous study showed a significant association with anorexia nervosa in a single locus analysis of rs1800955, while additional haplotype analysis showed a significant association at a four-locus haplotype including rs1800955 [144]. Previous associations were reported between alleles at -521 C/T and novelty seeking, extraversion, and drug use [145,146]. The rs1800955 polymorphism of *DRD4* was also proposed in the Genetic Addiction Risk Severity (GARS) test by Blum et al. This test identifies alleles that are proposed to impart vulnerability to addiction and makes an assessment of the degree of vulnerability of an individual to develop addictive behaviors [84]. This SNP has been associated with heroin addiction [139,147] and was reported to contribute to a “risk-taking phenotype” among skiers and snowboarders [148].

One of the aims of the present study was to investigate possible common genetic factors contributing to different substance use disorders and behavioral addictions. Based on the presented results, it appears that rs759364 of *FOXN3*, along with rs1549250 and rs2973033 of *GDNF*, may be non-specific genetic risk factors for various types of addictive behaviors. However, it should be noted that the results should be interpreted with caution due to the nature of the data (convenience sampling). Additionally, the measures of substance use behaviors are self-reported and were not assessed using structured scales. As adolescence and young adulthood are characterized by elevated levels of substance use, studies of larger samples with more diverse ages are also warranted. Furthermore, the candidate gene approach has limitations relating to *a priori* selection criteria and threshold levels, and the effect size values were also low. Therefore, the generalizability of the results has limitations and further studies are needed to confirm the role of these polymorphisms in addictions. Such studies may utilize multiple approaches, including haplotype analysis, polygenic risk scores, and genomic structural equation modeling. Nonetheless, as an initial study of genetic factors linked to substance use and behavioral addictions, it provides direction for future investigations to be conducted using independent samples.

## Figures and Tables

**Table 1 jpm-12-00690-t001:** Genetic association analysis results regarding addiction candidate genes and potentially addictive substance use.

			Substance Use											
Gene	dbSNP No.	Allele	Nicotine			Alcohol			Cannabis			Other Drugs		
			Non-Use (*n* = 3588)	Use (*n* = 1026)	*p*	Non-Use (*n* = 3538)	Use (*n* = 440)	*p*	Non-Use (*n* = 4204)	Use (*n* = 160)	*p*	Non-Use (*n* = 4088)	Ever Use (*n* = 1246)	*p*
**FOXN3**	**rs759364**	A	**30.8%**	**26.6%**	**0.016**	**30.0%**	**37.5%**	**0.0025 ***	30.4%	33.6%	0.425	30.1%	32.6%	0.103
		G	**69.2%**	**73.4%**		**70.0%**	**62.5%**		69.6%	66.4%		69.9%	67.4%	
GDNF	rs3096140	G	28.8%	28.2%	0.722	28.3%	25.7%	0.302	28.6%	24.1%	0.299	29.2%	29.0%	0.910
		A	71.2%	71.8%		71.7%	74.3%		71.4%	75.9%		70.8%	71.0%	
**GDNF**	**rs1549250**	C	**41.2%**	**45.1%**	**0.037**	40.8%	44.7%	0.139	41.0%	44.0%	0.491	**40.9%**	**46.0%**	**0.0029 ***
		A	**58.8%**	**54.9%**		59.2%	55.3%		59.0%	56.0%		**59.1%**	**54.0%**	
GDNF	rs2910702	C	23.9%	23.5%	0.808	23.3%	21.1%	0.321	23.4%	19.7%	0.327	23.5%	24.8%	0.386
		T	76.1%	76.5%		76.7%	78.9%		76.6%	80.3%		76.5%	75.2%	
GDNF	rs11111	C	15.5%	15.8%	0.848	15.4%	18.3%	0.125	15.3%	20.1%	0.123	15.6%	16.6%	0.419
		T	84.5%	84.2%		84.6%	81.7%		84.7%	79.9%		84.4%	83.4%	
**GDNF**	**rs2973033**	C	**27.8%**	**31.4%**	**0.040**	27.6%	32.2%	0.055	27.7%	32.6%	0.221	**27.3%**	**31.5%**	**0.0078 ***
		T	**72.2%**	**68.6%**		72.4%	67.8%		72.3%	67.4%		**72.7%**	**68.5%**	
GDNF	rs3812047	T	12.8%	12.3%	0.712	12.8%	11.4%	0.442	12.4%	8.1%	0.148	12.8%	12.9%	0.965
		C	87.2%	87.7%		87.2%	88.6%		87.6%	91.9%		87.2%	87.1%	
GDNF	rs1981844	C	28.5%	31.6%	0.082	28.3%	32.5%	0.098	28.3%	30.3%	0.622	28.2%	31.4%	0.049
		G	71.5%	68.4%		71.7%	67.5%		71.7%	69.7%		71.8%	68.6%	
**CNR1**	**rs806380**	G	33.7%	32.8%	0.620	33.5%	31.2%	0.347	32.9%	33.8%	0.823	**32.1%**	**36.4%**	**0.0077 ***
		A	66.3%	67.2%		66.5%	68.8%		67.1%	66.2%		**67.9%**	**63.6%**	
CNR1	rs2023239	C	18.2%	15.7%	0.081	17.8%	18.3%	0.835	17.9%	14.5%	0.300	18.4%	17.0%	0.288
		T	81.8%	84.3%		82.2%	81.7%		82.1%	85.5%		81.6%	83.0%	
DRD1	rs4532	C	37.7%	38.8%	0.575	37.1%	37.8%	0.806	36.9%	43.6%	0.111	**36.6%**	**40.5%**	**0.018**
		T	62.3%	61.3%		62.9%	62.3%		63.1%	56.4%		**63.4%**	**59.5%**	
DRD2	rs6277	G	46.8%	47.4%	0.766	47.8%	44.0%	0.147	46.6%	55.1%	**0.049**	47.0%	47.7%	0.674
		A	53.2%	52.6%		52.2%	56.0%		53.4%	44.9%		53.0%	52.3%	
**ANKK1**	**rs1800497**	A	18.8%	19.1%	0.829	19.5%	20.4%	0.671	18.6%	27.1%	**0.0113 ***	18.8%	20.5%	0.213
		G	81.2%	80.9%		80.5%	79.6%		81.4%	72.9%		81.2%	79.5%	
DRD3	rs6280	C	30.2%	30.8%	0.717	29.9%	33.8%	0.116	30.5%	32.4%	0.650	30.7%	29.9%	0.617
		T	69.8%	69.2%		70.1%	66.2%		69.5%	67.6%		69.3%	70.1%	
DRD4	rs1800955	C	46.0%	45.2%	0.691	45.9%	47.6%	0.554	45.2%	56.2%	**0.014**	45.9%	47.5%	0.367
		T	54.0%	54.8%		54.1%	52.4%		54.8%	43.8%		54.1%	52.5%	
CHRNA5/A3	rs16969968	A	35.2%	37.6%	0.188	34.0%	37.0%	0.230	35.8%	33.1%	0.520	35.1%	35.2%	0.973
		G	64.8%	62.4%		66.0%	63.0%		64.2%	66.9%		64.9%	64.8%	
CHRNA5/A3	rs1051730	A	35.6%	38.4%	0.124	34.9%	37.0%	0.422	36.5%	33.8%	0.517	35.8%	35.8%	0.984
		G	64.4%	61.6%		65.1%	63.0%		63.5%	66.2%		64.2%	64.2%	
CHRNB3	rs6474412	C	23.7%	20.6%	0.058	22.0%	22.3%	0.900	23.1%	26.5%	0.355	22.5%	23.0%	0.720
		T	76.3%	79.4%		78.0%	77.7%		76.9%	73.5%		77.5%	77.0%	
OPRM1	rs1799971	G	12.4%	13.9%	0.266	12.8%	12.5%	0.851	12.3%	12.1%	0.963	12.8%	12.8%	0.959
		A	87.6%	86.1%		87.2%	87.5%		87.7%	87.9%		87.2%	87.2%	
GABRA2	rs279858	C	39.5%	39.0%	0.785	39.7%	39.9%	0.924	40.7%	35.5%	0.224	40.0%	38.9%	0.506
		T	60.5%	61.0%		60.3%	60.1%		59.3%	64.5%		60.0%	61.1%	
TAS2R16	rs978739	C	33.7%	37.0%	0.060	33.0%	33.9%	0.719	34.2%	31.9%	0.570	34.1%	34.8%	0.662
		T	66.3%	63.0%		67.0%	66.1%		65.8%	68.1%		65.9%	65.2%	
FKBP5	rs1360780	T	28.4%	28.2%	0.916	28.6%	27.6%	0.677	27.9%	25.4%	0.535	27.7%	27.8%	0.927
		C	71.6%	71.8%		71.4%	72.4%		72.1%	74.6%		72.3%	72.2%	
FKBP5	rs4713916	A	28.0%	25.8%	0.197	27.9%	26.0%	0.442	27.5%	23.9%	0.356	27.3%	26.2%	0.488
		G	72.0%	74.2%		72.1%	74.0%		72.5%	76.1%		72.7%	73.8%	
ALDH2	rs886205	G	17.4%	18.3%	0.523	17.1%	19.3%	0.265	17.2%	21.4%	0.190	17.4%	18.2%	0.552
		A	82.6%	81.7%		82.9%	80.7%		82.8%	78.6%		82.6%	81.8%	
ALDH1B1	rs2073478	G	38.9%	36.3%	0.176	38.3%	38.0%	0.930	39.0%	37.5%	0.728	38.5%	39.0%	0.768
		T	61.1%	63.7%		61.7%	62.0%		61.0%	62.5%		61.5%	61.0%	
ADH1C	rs698	C	37.9%	40.5%	0.165	38.2%	37.4%	0.775	38.6%	34.4%	0.334	38.9%	37.3%	0.367
		T	62.1%	59.5%		61.8%	62.6%		61.4%	65.6%		61.1%	62.7%	
ADH1C	rs1693482	T	38.0%	40.3%	0.229	38.3%	37.0%	0.644	38.8%	33.8%	0.243	38.8%	37.5%	0.432
		C	62.0%	59.7%		61.7%	63.0%		61.2%	66.2%		61.2%	62.5%	
FAAH	rs324420	A	21.4%	24.4%	0.065	21.8%	22.4%	0.792	21.6%	21.6%	0.999	22.1%	20.1%	0.153
		C	78.6%	75.6%		78.2%	77.6%		78.4%	78.4%		77.9%	79.9%	
COMT	rs4680	G	47.6%	44.6%	0.125	47.6%	46.9%	0.778	47.7%	44.0%	0.403	47.0%	48.6%	0.361
		A	52.4%	55.4%		52.4%	53.1%		52.3%	56.0%		53.0%	51.4%	
WFS1	rs1046322	A	**10.0%**	**12.9%**	**0.013**	9.7%	11.9%	0.162	10.4%	11.4%	0.705	10.3%	10.2%	0.921
		G	**90.0%**	**87.1%**		90.3%	88.1%		89.6%	88.6%		89.7%	89.8%	
WFS1	rs9457	G	42.7%	44.8%	0.267	42.0%	46.7%	0.073	43.0%	50.7%	0.069	43.1%	43.4%	0.851
		C	57.3%	55.2%		58.0%	53.3%		57.0%	49.3%		56.9%	56.6%	
CALD1	rs3800737	C	31.0%	30.8%	0.883	30.6%	30.3%	0.903	30.8%	35.7%	0.221	29.7%	32.1%	0.128
		T	69.0%	69.2%		69.4%	69.7%		69.2%	64.3%		70.3%	67.9%	

Notes: Nominally significant associations are labeled in bold. * Significant after correction for multiple testing.

**Table 2 jpm-12-00690-t002:** Genetic association analysis results regarding addiction candidate genes and potentially addictive behaviors.

			Potentially Addictive Behaviors													
Gene	dbSNP No.	Allele	Internet Use (PIUQ)		Videogame Playing (POGQ)		Social Network Site Use (BSMAS)		Gambling (DSM IV MR J)		Exercise (EAI)		Trichotillomania (MGH-HPS)		Eating Disorders (SCOFF)	
			(*n* = 5936)	*p*	(*n* = 5698)	*p*	(*n* = 3452)	*p*	(*n* = 5916)	*p*	(*n* = 5916)	*p*	(*n* = 3250)	*p*	(*n* = 5978)	*p*
**FOXN3**	**rs759364**	A	**10.53**	**0.0003 ***	**16.11**	**0.003 ***	9.71	0.184	0.26	0.612	**12.18**	**0.003 ***	1.44	0.975	0.71	0.529
		G	**10.13**		**15.55**		9.52		0.25		**12.63**		1.45		0.73	
GDNF	rs3096140	G	10.36	0.109	15.72	0.438	9.75	0.109	0.24	0.474	**12.73**	**0.046**	1.38	0.894	0.71	0.449
		A	10.18		15.57		9.50		0.22		**12.41**		1.40		0.73	
GDNF	rs1549250	C	**10.13**	**0.027**	15.57	0.185	9.57	0.510	0.26	0.478	12.51	0.941	1.36	0.366	0.73	0.714
		A	**10.36**		15.80		9.66		0.24		12.52		1.49		0.72	
GDNF	rs2910702	C	10.27	0.937	15.78	0.597	9.80	0.185	0.27	0.383	12.58	0.624	1.35	0.491	0.71	0.742
		T	10.26		15.67		9.59		0.24		12.50		1.47		0.72	
GDNF	rs11111	C	10.07	0.104	15.43	0.179	9.35	0.078	0.24	0.636	12.69	0.235	1.26	0.270	0.72	0.875
		T	10.29		15.75		9.68		0.25		12.46		1.48		0.72	
GDNF	rs2973033	C	10.11	0.057	**15.41**	**0.031**	9.48	0.149	0.25	0.777	12.52	0.865	1.48	0.617	0.73	0.493
		T	10.33		**15.82**		9.70		0.25		12.49		1.40		0.71	
GDNF	rs3812047	T	10.35	0.482	15.67	0.982	9.73	0.495	0.20	0.167	12.66	0.400	1.45	0.967	0.73	0.805
		C	10.24		15.66		9.60		0.24		12.48		1.44		0.72	
GDNF	rs1981844	C	10.10	0.091	15.38	0.052	9.50	0.384	0.23	0.616	12.48	0.842	1.49	0.551	0.73	0.481
		G	10.30		15.76		9.64		0.25		12.51		1.39		0.71	
CNR1	rs806380	G	10.39	0.060	15.79	0.496	9.67	0.524	0.25	0.740	12.49	0.972	1.47	0.882	0.75	0.088
		A	10.19		15.66		9.58		0.24		12.49		1.45		0.71	
CNR1	rs2023239	C	10.37	0.284	15.63	0.674	9.56	0.823	0.20	0.080	12.41	0.656	1.43	0.943	0.72	0.905
		T	10.23		15.73		9.60		0.25		12.49		1.44		0.72	
DRD1	rs4532	C	10.21	0.394	15.71	0.861	9.53	0.427	0.25	0.620	12.58	0.341	1.41	0.837	0.72	0.979
		T	10.29		15.75		9.64		0.24		12.44		1.44		0.72	
DRD2	rs6277	G	10.17	0.158	15.59	0.239	9.57	0.664	0.27	0.058	12.45	0.538	1.37	0.329	0.71	0.473
		A	10.32		15.79		9.63		0.23		12.54		1.51		0.73	
ANKK1	rs1800497	A	10.21	0.686	15.83	0.536	9.87	0.051	0.24	0.917	12.46	0.855	1.33	0.441	0.73	0.772
		G	10.26		15.69		9.53		0.25		12.49		1.48		0.72	
DRD3	rs6280	C	10.20	0.420	15.80	0.638	9.45	0.094	0.28	0.080	12.41	0.413	1.39	0.654	**0.68**	**0.039**
		T	10.29		15.71		9.70		0.24		12.53		1.47		**0.73**	
**DRD4**	**rs1800955**	C	10.25	0.854	15.83	0.327	9.53	0.490	0.26	0.475	**12.31**	**0.042**	1.53	0.364	**0.67**	**0.002 ***
		T	10.23		15.65		9.63		0.24		**12.60**		1.38		**0.76**	
CHRNA5/A3	rs16969968	A	10.18	0.238	15.76	0.753	9.52	0.296	0.25	0.983	12.43	0.487	1.48	0.774	0.72	0.813
		G	10.31		15.71		9.66		0.25		12.53		1.43		0.73	
CHRNA5/A3	rs1051730	A	10.25	0.474	15.84	0.393	9.48	0.232	0.25	0.629	12.43	0.513	1.47	0.755	0.72	0.796
		G	10.32		15.68		9.65		0.24		12.52		1.42		0.72	
CHRNB3	rs6474412	C	**10.43**	**0.037**	15.61	0.719	9.51	0.510	0.26	0.709	12.52	0.723	1.39	0.638	0.74	0.429
		T	**10.18**		15.68		9.61		0.25		12.47		1.47		0.71	
OPRM1	rs1799971	G	10.25	0.899	15.60	0.571	9.41	0.303	0.21	0.186	12.56	0.691	1.51	0.727	**0.64**	**0.026**
		A	10.26		15.75		9.62		0.25		12.48		1.44		**0.73**	
GABRA2	rs279858	C	10.26	0.958	15.62	0.331	9.64	0.605	0.26	0.278	12.58	0.229	1.46	0.961	0.74	0.163
		T	10.26		15.79		9.57		0.24		12.41		1.45		0.71	
TAS2R16	rs978739	C	10.26	0.878	15.71	0.901	9.61	0.937	0.26	0.358	12.44	0.665	1.60	0.128	0.72	0.948
		T	10.27		15.74		9.60		0.24		12.51		1.36		0.72	
FKBP5	rs1360780	T	10.29	0.763	15.91	0.253	9.61	0.953	0.24	0.918	12.59	0.401	1.37	0.592	0.76	0.066
		C	10.26		15.68		9.62		0.24		12.45		1.46		0.70	
FKBP5	rs4713916	A	10.18	0.419	15.74	0.788	9.51	0.581	0.25	0.974	**12.73**	**0.031**	1.32	0.417	0.75	0.107
		G	10.27		15.69		9.60		0.25		**12.40**		1.46		0.70	
ALDH2	rs886205	G	10.09	0.165	15.71	0.999	9.45	0.377	0.27	0.420	12.56	0.617	1.42	0.951	0.74	0.310
		A	10.27		15.71		9.60		0.24		12.47		1.43		0.71	
ALDH1B1	rs2073478	G	10.29	0.578	15.68	0.929	9.64	0.561	0.23	0.451	12.44	0.999	1.45	0.927	0.74	0.276
		T	10.24		15.69		9.56		0.24		12.44		1.46		0.71	
ADH1C	rs698	C	10.35	0.285	15.90	0.077	9.58	0.844	0.24	0.397	**12.30**	**0.046**	1.42	0.830	0.70	0.276
		T	10.24		15.58		9.61		0.22		**12.58**		1.45		0.73	
ADH1C	rs1693482	T	10.33	0.254	15.88	0.128	9.58	0.892	0.26	0.466	**12.31**	**0.039**	1.42	0.742	0.71	0.542
		C	10.21		15.61		9.60		0.24		**12.60**		1.47		0.72	
FAAH	rs324420	A	10.12	0.171	15.56	0.361	9.58	0.834	0.24	0.326	12.43	0.612	1.63	0.138	0.72	0.832
		C	10.29		15.75		9.61		0.26		12.51		1.35		0.72	
COMT	rs4680	G	10.19	0.262	**15.51**	**0.039**	9.58	0.593	0.24	0.689	12.40	0.155	**1.29**	**0.026**	0.72	0.825
		A	10.31		**15.87**		9.65		0.25		12.60		**1.63**		0.72	
WFS1	rs1046322	A	10.05	0.134	15.50	0.395	9.57	0.868	0.25	0.841	12.22	0.198	1.37	0.740	0.73	0.785
		G	10.29		15.75		9.61		0.25		12.51		1.46		0.72	
WFS1	rs9457	G	10.33	0.229	15.78	0.559	9.69	0.261	0.26	0.186	12.61	0.144	1.52	0.356	0.74	0.170
		C	10.20		15.67		9.53		0.24		12.40		1.39		0.70	
CALD1	rs3800737	C	10.29	0.638	15.49	0.060	9.63	0.690	0.23	0.188	12.49	0.985	1.38	0.547	0.73	0.557
		T	10.24		15.84		9.58		0.26		12.49		1.48		0.71	

Notes. Nominally significant associations are labeled in bold. * Significant after correction for multiple testing.

## Data Availability

The data analyzed during the current study are available in the OSF repository, https://osf.io/nks43/?view_only=ebcc2169bcb4493191f64b45426b9f91; https://doi.org/10.17605/OSF.IO/NKS43 (accessed on 6 February 2022).

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
