# Peer review of "FOXN3 and GDNF Polymorphisms as Common Genetic Factors of Substance Use and Addictive Behaviors"

_jpm, 2022, doi:10.3390/jpm12050690_

Round 1

Reviewer 1 Report

The paper is set up correctly, the statistics sufficient. There are a couple of concerns that should be considered

The first point: was anonymity guaranteed for the data collected in the questionnaire, in order to avoid sampling bias? Please make this clear.

The second refers to what has already been stated on p. 16 "results should be interpreted with caution due to the nature of the data (convenience sampling). Additionally, the measures of substance use behaviors are self-reported and were not assessed using structured scales." This sentence that should be underlined and a brief mention should be reported in the abstract.

Author Response

We appreciate the feedback provided by the Reviewer to help improve our manuscript. We have incorporated the suggested changes into the manuscript.

Response1: Yes, participation was anonymous. The following has been incorporated in the Materials and Methods section of the manuscript: ‘Participation was anonymous, questionnaire data and DNA information were paired by a unique identification number of the participants.’

Response2: As the reviewer suggested, the convenience sampling technique and the lack of structured scales for substance use behaviors were highlighted in the abstract as follows: ‘Due to limitations (e.g. convenience sampling, lack of structured scales for substance use) further studies are needed.’

Reviewer 2 Report

Vereczkei et al. analyzed the genetic overlap of various substance use, addictive, and other compulsive behaviors. They conducted an association analysis of 32 SNPs and four potentially addictive substances and seven potentially addictive or compulsive behaviors in a group of 3003 participants. The results of the study indicate that FOXN3 and GDNF genes are involved in both substance and behavioral addictions. The results of this study are highly significant.

However, there are some minor issues:

  1. Introduction/ Material and Methods. More information is needed about the genes and specific variants selected for analysis. Please provide citations. Lines:138-140 and 185-193.
  2. In paragraph describing GDNF gene significant variants and their role in addictions please provide information on both SNPs – location in the gene, impact on peptide if in exon and functional rationale.

Author Response

We thank the reviewer for valuing the importance of the results of our current study. We appreciate the Reviewer’s comments, the indicated changes have been incorporated into the revised version of the manuscript.

Response1: According to the Reviewer’s request, the following citations regarding the SNP selection have been included in the SNP selection criteria paragraph of the Materials and Methods section.

Agrawal, A., & Lynskey, M. T. (2009). Candidate genes for cannabis use disorders: findings, challenges and directions. Addiction, 104(4), 518-532. doi: 10.1111/j.1360-0443.2009.02504.x

Bierut, L. J. (2009). Nicotine dependence and genetic variation in the nicotinic receptors. Drug Alcohol Depend, 104 Suppl 1, S64-69. doi: 10.1016/j.drugalcdep.2009.06.003

Hinrichs, A. L., Wang, J. C., Bufe, B., Kwon, J. M., Budde, J., Allen, R., . . . Goate, A. M. (2006). Functional variant in a bitter-taste receptor (hTAS2R16) influences risk of alcohol dependence. Am J Hum Genet, 78(1), 103-111. doi: 10.1086/499253

Husemoen, L. L., Fenger, M., Friedrich, N., Tolstrup, J. S., Beenfeldt Fredriksen, S., & Linneberg, A. (2008). The association of ADH and ALDH gene variants with alcohol drinking habits and cardiovascular disease risk factors. Alcohol Clin Exp Res, 32(11), 1984-1991. doi: 10.1111/j.1530-0277.2008.00780.x

Response2: The gene locations of the GDNF SNPs have been included in the Discussion section as the reviewer suggested. The rs1549250 is an intronic variant of the gene, while rs2973033 is located in the 5’ untranslated region. To date, associations of GDNF SNPs with behavioral addictions (e.g. gambling) have been shown, but no functional consequences have been published on these variants.